# OpenReview forum: "AEGIS2.0: A Diverse AI Safety Dataset and Risks Taxonomy for Alignment of LLM Guardrails"
_NeurIPS.cc/2024/Workshop/SafeGenAi — SafeGenAi Poster_

### Official Review · Reviewer_DKip · 2024-10-08
**complete, clear and detailed research**

**Rating:** 7
**Confidence:** 3

**Review:**

Advantage points:
1. complete and detailed classification: this paper proposes a comprehensive and detailed security risk classification method, which provides important reference for the security assessment of LLM and has strong practical application value.
2. jury system: combining the advantages of manual annotation and LLM,  provided a large-scale and diverse dataset with high value in application.
3. clear and detailed experiments setting, easy to reproduce

Weakness:
1. lack of open source, So I can't be sure if the claimed effect of the paper is useful

---

### Official Review · Reviewer_aXWv · 2024-10-09
**Comprehensive taxonomy and strong experimental results but requires enhancements in literature analysis, implications, and limitations within main text.**

**Rating:** 7
**Confidence:** 5

**Review:**

Title: AEGIS2.0: A Diverse AI Safety Dataset and Risks Taxonomy for Alignment of LLM Guardrails

Strengths:
1.	Comprehensive dataset with a well-defined taxonomy.
2.	Strong experimental results demonstrating model efficacy.
3.	Clear alignment with real-world safety concerns in LLMs.
4.	Thoughtful ethical considerations.

Weaknesses:
1.	Some areas lack depth, particularly in discussing implications and limitations within the main text.
2.	The literature review could provide a more critical perspective on key works such as Abdelkader et al. (2024), Ravi, K., & Vela (2024), and Ravi et al. (2023), which offer important insights into AI safety guards, AI threat datasets, and risk taxonomies for detection mechanisms relevant to this paper.

Recommendations:
To enhance the impact of the paper, the authors should consider:
1.	Enhancing the related works section to provide a more critical analysis.
2.	Offering a more detailed justification for the chosen categories in the taxonomy.
3.	Expanding discussions on the implications of findings for both research and industry.
4.	Thoroughly addressing potential limitations and future research avenues.

Conclusion:
Overall, this work represents a significant advancement in the development of a dataset for AI safety, effectively tackling critical issues in large language models. By refining certain sections and incorporating the suggested recommendations, the authors can strengthen the paper's contributions and applicability in both academic and practical contexts. As AI continues to evolve, the insights gained from this research can inform the creation of more robust defenses against cyber threats, ultimately fostering a safer digital environment.

References:
Abdelkader, H., Abdelrazek, M., Barnett, S., Schneider, J. G., Rani, P., & Vasa, R. (2024, April). ML-On-Rails: Safeguarding Machine Learning Models in Software Systems—A Case Study. In Proceedings of the IEEE/ACM 3rd International Conference on AI Engineering-Software Engineering for AI (pp. 178-183).
Ravi, K., & Vela, A. E. (2024). Comprehensive dataset of user-submitted articles with ideological and extreme bias from Reddit. Data in Brief, 56, 110849.
Ravi, K., Vela, A. E., Jenaway, E., & Windisch, S. (2023, December). Exploring Multi-Level Threats in Telegram Data with AI-Human Annotation: A Preliminary Study. In 2023 International Conference on Machine Learning and Applications (ICMLA) (pp. 1520-1527). IEEE.

---

### Official Review · Reviewer_Z1oS · 2024-10-10
**Safety Data for Training**

**Rating:** 6
**Confidence:** 3

**Review:**

The authors introduce a dataset intended to help train guard models against a new taxonomy of harm categories. They run a variety of experiments to evaluate the dataset's effectiveness for its intended purpose. My enthusiasm for the paper is tempered by the extensive use of LLMs in the annotation of the dataset. It is difficult to isolate what impacts this may have on downstream applications since the guard models will be tuned subject to the residuals of the annotation LLMs.

Reviews are expedited to ensure authors have enough time to apply for a visa. Consequently, I am reviewing more at the surface level consistent with a non-archival workshop. I may get things wrong, but I spent as much time with the paper as is necessary for me to render a decision that would hold up under much greater scrutiny. In this case, that is ensuring the paper has the hallmarks of being well supported. NeurIPS workshops are also more influential than many conference proceedings, so I am also being selective.